# Genetic Parameters for Growth and Kid Survival of Indigenous Goat under Smallholding System of Burundi

**DOI:** 10.3390/ani10010135

**Published:** 2020-01-15

**Authors:** Manirakiza Josiane, Hatungumukama Gilbert, Detilleux Johann

**Affiliations:** 1Department of Animal Health and Productions, Faculty of Agronomy and Bioengineering, University of Burundi, Bujumbura B.P. 2940, Burundi; hatungumukama@yahoo.com; 2Fundamental and Applied Research for Animals and Health, Faculty of Veterinary Medicine, University of Liège, 6 Avenue de Cureghem, 4000 Liège, Belgium; jdetilleux@uliege.be

**Keywords:** body weight, heritability, repeatability, genetic correlations, Bayesian approach, survival analysis

## Abstract

**Simple Summary:**

Goats play a key multifunctional role in food security and poverty alleviation for small farmers in many less developed countries. Unfortunately, the productivity level of these goats is low. Among the alternatives proposed to overcome this situation, one is to establish a phenotypic breeding program with the support of community breeding organizations. In this study, genetic parameters were estimated for the growth, conformation, and survival of 1538 young goats raised by small farmers in Burundi organized in farmer field schools. Overall, the results suggest that phenotypic selection of growth and conformation traits would be possible if data recording and animal management were improved. On the other hand, efforts to improve survival should focus on improving the environmental conditions in which kids are raised. The role of community breeding organizations and animal health workers is therefore essential to disseminate breeding techniques and methods that optimize animal production and health.

**Abstract:**

The goal of this study was to estimate genetic parameters for the growth, conformation, and survival of goat kids raised in smallholder farming systems in Burundi. To do this, measurements were taken on live weight, thoracic perimeter, length, and height at birth (n = 1538 animals), at 3 months (n = 1270 animals), at 6 months (n = 992 animals), at 9 months (n = 787 animals), and at 12 months (n = 705 animals). Kids were born between 2016 and 2019, from 645 dams and 106 bucks. Three bivariate animal models were used to estimate genetic parameters of body weight and conformation measurements as potential indicators of this weight. According to the measure, heritability was estimated between 15 and 17% and genetic correlations between 65 and 79%. An accelerated failure time animal model was used to estimate the heritability of survival for kids under one year, adjusted for birth weight. Goat survival was significantly prolonged by 0.64 days per kilogram of birth weight. The estimated heritability for this trait was 2%. Overall, these results suggest that a selection program could be implemented to improve animal growth, either directly on weight or indirectly on conformational traits. At the same time, efforts need to be made to improve rearing conditions to increase the survival of kids.

## 1. Introduction

Goats play a key multifunctional role in food security and poverty alleviation of smallholder farmers in many less developed countries such as Burundi. Native goats, the most important breed, were considered the most suitable species for harsh environments [1] but they are not productive enough to ensure food security and the well-being of goat farmers [2]. One reason is that native breeds have been naturally selected to be adapted to their marginal environment, but not to increase their level of performance [3]. Another reason is that these goats are considered a source of income for urgent and regular needs, which leads to negative selection by selling fast-growing kids [4].

Given their low level of productivity, it may be feared that native goats will vanish [5] so different genetic interventions aimed at improving their productivity level have been proposed. We recognize that productivity could be boosted by improving animal management [6], but this requires skills that farmers do not necessarily possess [7]. In addition, genetic improvement is inherently cumulative from one generation to the next and this is not the case for animal management. 

One genetic intervention has been to cross them with imported breeds. But a consequence of this crossbreeding scheme is that the local, old, and well-adapted breed could be left behind, along with its unique genetic makeup. 

Another alternative would be to set up a genotypic breeding program whose goals would be to improve goat productivity. To do this, accurate information on pedigree and performance is needed. This information is often not available because of the many constraints on small-scale farming systems in the least developed countries. These include the high illiteracy rate among smallholders, the lack of animal identification and pedigree recording systems, the non-existence of institutional frameworks, and the inadequacy of village-level organizations to ensure an effective participation of the farmers in breeding programs [8,9,10]. Thus, genotypic breeding programs have been implemented in stations, but the disadvantages are numerous. These include the incompatibility between the environments in the station and the conditions in the village, the financial and technical problems due to the lack of long-term commitment of institutions in developing small ruminants breeding, and the non-participation of farmers in the design and implementation of these breeding programs [11]. 

Another solution would be to set up a phenotypic selection program directly on the farm. In such cases, community-breeding organizations have an important role to play. In fact, they have been set up in many developing countries to support programs to improve the performance of small ruminants [11,12,13]. Among others, the role of such organizations is to overcome the challenge of the small organizational capacity of smallholders to collect accurate data. One point to consider in phenotypic selection is that it would be effective only if the traits to be improved had “good genetic parameters”, including heritability or genetic correlation. For example, if the heritability of a trait is high, the effects on the environment would be low and it would not be necessary to obtain information on pedigree and genotypes for genetic progress to be effective [14]. Efficiency of phenotypic selection can also be improved by indirect selection based on the correlation of traits because traits that are cheap or easy to measure and that have favorable genetic correlations with economically important traits that are more difficult or expensive to measure can be utilized as indicator traits.

This is the option taken in this study. We have collected information on goat characteristics in pilot farmer associations called farmer field school (FFS), as they help producers build their technical capacity. Researchers and local veterinary services followed the FFS and collected information on goat performance and pedigree. Selection objectives and criteria to be improved in the phenotypic selection program have been identified after surveying the farmers [7]. This survey showed that mortality and slow growth rates were the main factors limiting the profitability of goats farming, as they reduced the number and value of marketable kids. In addition, these factors had a negative impact on genetic improvement by reducing the size of the breeding nucleus, as well as the amount of data to estimate genetic parameters.

Therefore, this study aims to estimate the genetic parameters of growth and mortality rates of local goats raised in smallholder Burundian farming systems to determine whether phenotypic selection would be relevant to these traits.

## 2. Materials and Methods

### 2.1. Study Area 

The study area included the provinces of Gitega and Rutana. Gitega province is located in the humid central uplands and Rutana province is located in the dry eastern lowlands. The geoclimatic and management characteristics of goats in these provinces are summarized in Table 1. Two communes were identified in each province: Nyarusange and Ryansoro in the Gitega province, and Rutana and Bukemba in the Rutana province. In each of these communes, two villages and 15 farmers per village were trained to form an FFS.

### 2.2. Field Data

Each animal has been identified with an ear tag number and each event (e.g., birth, abortion, diseases, mortality) has been recorded. The data covered a period of 3 years, between 2016 and 2019. Dams remained in the flock until death while the bucks were replaced when the first offspring of their son were old enough to mate. Breeding males were traded between farmers while the others were sold.

For each kid, a record included the parents’ identity codes, its sex, date and type of birth, dam’s parity, herd code, and performance. Body weight (BW), chest girth (CG), body length (BL), and height at withers (HW) were recorded at birth, 3 months, 6 months, 9 months, and 12 months. The BW is the selection criterion, and CG, BL, and HW are indicator traits. Measurements at birth had to be recorded within 3 days of birth. Measurements were taken (with a mobile scale and measuring tape) by trained community animal health workers in each village. Data has been regularly recorded in a central data set. Survival time was measured as the number of days alive over a one-year period since birth. Usually, one year is age at first service or optimal sale, as reported by farmers. Animals who have not experienced death during this period were right-censored (unknown survival time, but at least 360 days).

During the follow-up period, a total of 1538 kids were born to 645 dams and 106 sires with 1538, 1270, 992, 787, and 705 records at birth, 3 months, 6 months, 9 months, and 12 months, respectively. About 1149 records on length of survival were gathered. 

### 2.3. Data Analysis

#### 2.3.1. Body Weight and Linear Measurements

After computing preliminary descriptive statistics, records were analyzed assuming three bivariate repeatability-maternal animal models, one for BW and CG, one for BW and BL, and one for BW and HW:
$$\begin{array}{c} \left[\begin{array}{c} y_{1}\\ y_{1} \end{array}\right]=\left[\begin{array}{c} X_{1}\;\;0\\ 0\;\;X_{2} \end{array}\right]\left[\begin{array}{c} b_{1}\\ b_{2} \end{array}\right]+\left[\begin{array}{c} \;Z_{a1}\;\;\;\;0\\ \;0\;\;\;\;Z_{a2} \end{array}\right]\left[\begin{array}{c} a_{1}\\ a_{2} \end{array}\right]+\left[\begin{array}{c} \;Z_{m1}\;\;\;\;0\\ 0\;\;\;\;Z_{m2} \end{array}\right]\left[\begin{array}{c} m_{1}\\ m_{2} \end{array}\right]+\left[\begin{array}{c} Z_{p1}\;\;\;\;0\\ 0\;\;\;\;Z_{p2} \end{array}\right]\left[\begin{array}{c} p_{1}\\ p_{2} \end{array}\right]\\ +\left[\begin{array}{c} Z_{c1}\;\;\;0\\ 0\;\;\;\;Z_{c2} \end{array}\right]\left[\begin{array}{c} c_{1}\\ c_{2} \end{array}\right]+\left[\begin{array}{c} e_{1}\\ e_{2} \end{array}\right] \end{array}$$
where y_1_ is the vector of observations for BW and y_2_ is the vector of observations for CG, BL, or HW; b_1_ and b_2_ are the corresponding vectors of fixed effects; a_1_ and a_2_ are the corresponding vectors of additive genetic effects; m_1_ and m_2_ are the vectors of maternal effects; p_1_ and p_2_ are the vectors of permanent environmental effects; c_1_ and c_2_ are the vectors of common environmental effects; e_1_ and e_2_ are the vectors of random residuals; and X_i_, Z_ai_, Z_pi_, Z_mi_, and Z_ci_ are the incidence matrices relating y_i_ to the corresponding vectors (i = 1 et 2). The fixed effects included in all three models were: age (n classes = 5), sex (n classes = 2), parity (n classes = 3), and type of birth (n classes = 2), as well as all interactions. Furthermore, random components were assumed normally distributed: a ~ N (0, A σ^2^_a_), m ~ N (0, I σ^2^_m_), pe ~ N (0, I σ^2^_p_), ce ~ N (0, I σ^2^_c_), and e ~ N (0, I σ^2^_e_) where σ^2^_a_, σ^2^_m_, σ^2^_p_, σ^2^_c_ and σ^2^_e_ are the variances components and A is the additive genetic relationship matrix. 

#### 2.3.2. Survival Analysis

An accelerated failure time (AFT) animal model [15,16,17] was used to estimate the effects of risk potential factors affecting survival time. The model is:log (ST) = X b + Z_a_ a + Z_m_ m + Z_c_ c + e
where ST represents the vector of survival times; log is the logarithm; b is the vector of fixed effect of birth weight; a is the vector of additive genetic effects; m is the vector of maternal effects; c is the vector of common environmental effect (i.e., a vector of random herd, year, and season of death); e is the vector of random residuals; and X, Z_a_, Z_c_, and Z_m_, are incidence matrices relating y to corresponding vectors. Exponential distributions were assumed for all random components where σ^2^_a_, σ^2^_m_, σ^2^_c_, and σ^2^_e_ are the variances components and A is the additive genetic relationship matrix. 

#### 2.3.3. Bayesian Estimation

For both models, variance components and genetic parameters were estimated using a Bayesian approach and with the MCMCglmm package-R [18,19]. Inverse-gamma prior distributions with different values for the shape and scale parameters were chosen as priors for all variance matrices [20,21]. Number of MCMC iterations was set at 100,000, burn-in at 10,000, and thinning interval at 10. Convergence to the target distribution was checked through visual inspection of all trace and density plots.

Output of the iterations were used to calculate the mean, median, and highest posterior density (HPD) interval for each variance/covariance components [18,21,22]. Phenotypic variances (σ^2^_ph_) of body weights and linear measurements were computed as: σ^2^_ph_ = σ^2^_a_ + σ^2^_m_ + σ^2^_p_ + σ^2^_c_ + σ^2^_e_. Then, heritability (h^2^) and repeatability (r) were computed as: h^2^ = σ^2^_a_/σ^2^_ph_ and r = (σ^2^_a_ + σ^2^_p_)/σ^2^_ph_. Genetic correlations between BW and one of the indicator trait (CG, BL, or HW) were computed with the following model: r_g_ = g_12_/(g_11_*g_22_)^1/2^ where g_12_ is the additive genetic covariance between the two traits, and g_11_ and g_22_ are the additive genetic variance for trait 1 and 2, respectively.

## 3. Results

Overall means BW (±standard deviation) at birth, 3 months, 6 months, 9 months, and 12 months were 2.1 ± 0.5, 6.4 ± 1.6, 9.0 ± 2.2, 11.4 ± 2.5, and 13.6 ± 2.9 kg, respectively. Phenotypic correlations between BW and CG, BW and BL, and between BW and HW were 0.95, 0.94, and 0.93, respectively. Mean litter size at birth was 1.4 kids, slightly smaller in the Gitega (1.2) than in the (1.6) Rutana province. Mortality rate up to one year was estimated at 31.6% and was higher among twins (42%) than among kids born alone (24.2%).

Results of the statistical analyses are given in Table 2: Posterior means, medians, and limits of HPD intervals for h^2^, r, genetic correlations, and all variance components. There appears to be a tendency for estimates of σ^2^_m_ to be smaller than the other variance components, especially for BW, CG, BL, and HW. Inversely, estimates of σ^2^_e_ are highest for CG, BL, and HW and estimates of σ^2^_c_ are highest for ST. Heritability estimates are very low for BL and log (ST) and close to 15% for BW, CG, and HW. Genetic correlations are between 60 and 70%. 

Age, sex, and type of birth significantly influenced body weights. For example, posterior BW mean of twins was 1 kg lower than posterior BW mean of singles (0.7 and 1.3 kg for the lowest and highest HPDP limits). Posterior BW mean of males was 0.58 kg heavier than females (0.35 and 0.83 for the lowest and highest HPDP limits). Concerning ST, the time ratio to death is expected to decrease significantly by 0.64 day per one kg increase in birth BW.

## 4. Discussion

The main objective of this study was to obtain BW and ST genetic parameters to confirm whether phenotypic selection of these traits would be possible in smallholding farms regrouped in FFS. Our results are more or less consistent with estimates published in the literature. For example, our BW h^2^ estimate is within the range (0.09 to 0.47) reported by several authors from BW measured between birth and yearling age in different goat breeds [23,24,25,26]. However, it is higher than the one reported by [27,28,29]. Similarly, our ST h^2^ estimate is in the range reported by [30,31]. 

Given that estimates of h^2^ for growth related traits were low to moderate, on-farm phenotypic selection to improve these traits is thought to be not very effective in the current situation. This is because the expected response to selection, i.e., the gain achieved by mating the selected parents, is directly related to h^2^ [32]. But the situation could be improved with better recording and management. Indeed, h^2^ values were not caused by a paucity of σ^2^_a_ but by great residual variances. For example, median estimates of σ^2^_a_ for BW is 0.58 kg^2^, much lower than the estimate for σ^2^_e_ at 1.57 kg^2^. A similar disparity is also noticeable for CG, BL, and HW. Moreover, estimates of repeatability, which is often considered as the upper limit to heritability [33], were close to 30% for BW, CG, BL, and HW. So, given that σ^2^_a_ is not negligible and that repeatability is 30%, there may be some genetic variation between animals and it would be possible to select animals as parents of the next generation based on their phenotypic values [34] for growth. 

Other factors, such as the shallow depth of the pedigree, may also explain the low h^2^ estimates. Indeed, missing parentage data have been shown to influence h^2^ estimates in domestic and wild species [35]. Another element concerns human errors when recording mating details and phenotypic information, despite our efforts to verify the data. This is a genuine problem, as wrong and missing pedigree information leads to a downward bias of both heritability estimates and genetic covariances [36] and reduces genetic gain [37]. We included maternal environmental effects in our models as they contribute to offspring phenotypic variation [38,39,40] but covariance can also exist between direct additive and maternal genetic effects. If this covariance is negative, a gene with a positive effect on an offspring trait may have a negative effect on maternal performance for that trait and this will may act to maintain genetic variance. Then, h^2^ is not necessarily a useful measure of a trait’s potential to evolve [41]. Finally, possible allelic interactions within loci (dominance) and between loci (epistasis) may also explain parts of the total genetic variation [42] and should not have been neglected. We acknowledge these limits but our data were unfortunately not sufficient to allow good estimates of all these parameters simultaneously. A more in-depth analysis must be done to check our results.

Estimates of genetic correlations between BW and CG, BL and HW are high, suggesting these indicator traits could be used to indirectly select for better growth. As it has been reported elsewhere [22,43], the most important correlation was found between BW and CG (0.79). This may be because GC and BW are both tissue-related measurements while BL and HW are skeletal-related measurements [44]. Whatever the cause, this observation is important because goat prices in the market are systematically decided after visual observation of the animal’s conformation, and less according to BW. Estimates of repeatability of measures of BW, CG, BL, and HW across ages are all around 30%. As a measure a consistency of these measures, it indicates that multiple measurements of BW, CG, BL, and HW are necessary to evaluate accurately the animal over its lifetime [33]. For example, knowing the weight at an early age would less predict the adult weight. 

Concerning ST, h^2^ is null and common environmental variance is very high. Therefore, the expected response to selection for this trait would be almost null and efforts should focus on improving the environmental conditions in which kids are raised, especially as the number of days of survival increases as birth weight increases. This is a great challenge because veterinary surveillance systems are weak. As an illustrative example, the outbreak of small ruminant pest decimated more than 4000 goats herds in 2018 [45]. A meta-analysis shows that animal diseases kill about 18% of livestock in low-income countries [46]. At an institutional level, some authors [47] propose various recommendations to alleviate the animal health situation in Africa, among which the role of FFS and community-based animal health workers to disseminate farming techniques and methods that optimize animal production and health [48,49]. 

Their role is also essential to ensure the sustainability of breeding programs, as these require a long-term commitment from all stakeholders [50]. In this study, we also suggest that the on-farm phenotypic selection program be accompanied by a genotypic selection program conducted under well-controlled conditions in a research center. Selected elite bucks could be distributed to the goat population of FFS members. However, a long-term link between the FFS and the center, as well as the sources of funding, must be well studied.

## 5. Conclusions

Results of estimates genetic parameters in this study indicated that a modest genetic progress could be achieved by on farm phenotypic selection of indigenous goats with better weight and conformation. This phenotypic selection should be accompanied by efforts on improving recording and management. On the other hand, response to selection for kid survival would be almost null and efforts should focus on improving the environmental conditions in which kids are raised.

## Figures and Tables

**Table 1 animals-10-00135-t001:** Characteristics of the climate and goat management of the study areas.

Province	Altitude (m)	Annual Temperature (°C)	Annual Rainfall (mm)	Dry Season (Months)	Goat Management
Gitega	1350–2000	17–25	1200–1500	From June to September or October	Stall-feeding with forage crops (*Pennisetumm purpureum*, *Trypsacum laxum*, and *Setaria sphacelata*) and crops by-products
Rutana	1125–1400	22–28	900–1200	From May to November	Free grazing complemented with crop by-products

**Table 2 animals-10-00135-t002:** Mean, median, and highest posterior density (HPD) lowest and highest limits for genetic parameters of body weights, chest girth, body length, height at whither, and survival up to 12 months: additive (σ^2^_a_), maternal (σ^2^_m_), permanent environmental (σ^2^_p_), and common environmental (σ^2^_c_) variance components; repeatability (r) and heritability (h^2^) and genetic correlations between body weight and chest girth (BW_CG), body weight and body length (BW_BL), and between body weight and height at wither (BW_HW).

Trait	Genetic Parameters	Mean	Median	Lowest	Highest
Body weight	σ^2^_a_ (kg^2^)	0.58	0.57	0.37	0.79
	σ^2^_m_ (kg^2^)	0.22	0.23	0.09	0.38
	σ^2^_p_ (kg^2^)	0.55	0.54	0.37	0.70
	σ^2^_c_ (kg^2^)	0.41	0.42	0.33	0.53
	σ^2^_e_ (kg^2^)	1.57	1.57	1.5	1.61
	h^2^	0.17	0.17	0.11	0.23
	r	0.33	0.33	0.29	0.38
Chest girth	σ^2^_a_ (cm^2^)	2.56	2.53	1.44	3.65
	σ^2^_m_ (cm^2^)	1.5	1.49	0.76	2.2
	σ^2^_p_ (cm^2^)	2.08	2.08	1.23	2.88
	σ^2^_c_ (cm^2^)	1.15	1.15	0.8	1.5
	σ^2^_e_ (cm^2^)	7.95	7.95	7.56	8.31
	h^2^	0.16	0.16	0.09	0.24
	r	0.30	0.30	0.26	0.35
Body length	σ^2^_a_ (cm^2^)	0.59	0.54	0.0004	1.49
	σ^2^_m_ (cm^2^)	1.00	1.00	0.39	1.59
	σ^2^_p_ (cm^2^)	3.10	3.12	2.31	3.86
	σ^2^_c_ (cm^2^)	1.19	1.19	0.86	1.54
	σ^2^_e_ (cm^2^)	7.00	6.99	6.67	7.34
	h^2^	0.05	0.04	0.0003	0.11
	r	0.29	0.28	0.23	0.33
Height at wither	σ^2^_a_ (cm^2^)	1.65	1.61	0.86	2.5
	σ^2^_m_ (cm^2^)	1.37	1.36	0.79	1.94
	σ^2^_p_ (cm^2^)	1.77	1.78	1.14	2.41
	σ^2^_c_ (cm^2^)	1.12	1.11	0.83	1.44
	σ^2^_e_ (cm^2^)	5.96	5.96	5.68	6.25
	h^2^	0.13	0.14	0.07	0.21
	r	0.29	0.29	0.24	0.34
Kid survival	σ^2^_a_ (days^2^)	0.05	0.06	0.03	0.10
	σ^2^_m_ (days^2^)	0.05	0.06	0.03	0.10
	σ^2^_c_ (days^2^)	1.78	1.81	1.35	2.32
	σ^2^_e_ (days^2^)	0.05	0.05	0.03	0.09
	h^2^	0.02	0.02	0.01	0.04
Correlation	BW_CG	0.79	0.75	0.52	0.90
	BW_BL	0.65	0.61	0.24	0.85
	BW_HW	0.74	0.72	0.47	0.87

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
