# Peer review of "Genetic Parameters for Growth and Kid Survival of Indigenous Goat under Smallholding System of Burundi"

_animals, 2020, doi:10.3390/ani10010135_

Round 1

Reviewer 1 Report

This is a very good, interesting and well-written article which provides interesting data and their clear interpretation. What is missing is the Simple Summary but it’s the Editor’s discretion to decide if it’s necessary. Except that I have only minor comments:

Line 100: should be “selection criterion”

Line 108: insert “,” after month

Line 154: add a decimal place after 9 – 9.0

Line 223: better to write: “At institutional level, some authors [47] propose…”

Author Response

Thank you very much for the review. We appreciate greatly the speed of the review process. We accept all comments of the reviewer and adapt the manuscript accordingly. All changes in the manuscript are highlighted in yellow.

Reviewer 2 Report

Manuscript Animals-651014, entitled “Genetic Parameters for Growth and Kid Survival of Indigenous Goat Under Smallholding System of Burundi”

Recommendation:       The above paper is not suitable for publication in its present form.

General Comments:

The article provides useful information about the genetic parameters for growth of indigenous goat under smallholding system in Burundi. Although the experiment is appropriately designed and implemented, there are a lot of grammar, stylistic and syntax errors. In some cases, these errors negatively influence the understanding of the text.

My main concern is that the results of this study are somehow expected. It is already known that growth parameters are positively correlated with birth weight and survival is mostly influenced by environmental factors. The novelty of this study could be possibly relied on the indigenous goat breed and on the area of the study (Burundi).

Some minor points should also be corrected.

Specific comments

L13-14: “…farming systems in Burundi. Live weight…”

L16: “…and at 12 months (n = 705 animals) were recorded. Kids were…”

L18: “According to the analysis…”

L21-22: This statement is somehow bizarre. Only 0.64 days?

L24-25: “…efforts should be made on improving rearing conditions…”

L34-35: “…their marginal environment, but their level of performance is not comparable to that of the high-yielded goats [3]. Another reason…”

L37-38: “Due to their low level of productivity, there is a fear (?) that native goats will be vanished [5] so different genetic interventions that are aimed at…”

L39: “…but this improvement requires…”

L41: “in contrast with” instead of “and this is not the case for”

L42: “One genetic intervention could be to cross native goats with imported…”

L46: “necessary” instead of “needed”

L52: “variability” instead of “incompatibility”

L57-58: “In this case…”

L58-59: “In fact, such programs are already implemented in many developing countries to improve the performance…”

L61: “in the collection of” instead of “to collect”

L64: “…the effects of the environment…”

L70: “examined” instead of “taken”

L71: “…help producers on building their…”

L80: “…farming systems and to determine…”

L92: “Each animal was identified…”

L93: “…diseases, mortality) was recorded.”

L97: Please delete “a record included”

L98: “…and performance were recorded. Body weight…” What do you mean by “performance”?

L99: “…at withers (HW) were also measured at birth…”

L99: “The BW was the selection criterion and CG, BL and HW were indicator traits.”

L101: “implemented” instead of “taken”

L104-105: “Usually, one year is the age of the first service or optimal sale, as reported by farmers. Animals that were not dead during this period…”

L140: “…approach by the MCMCglmm…”

L141: “selected” instead of “chosen”

L153: “Overall means for BW…”

L156: “…than in the Rutana (1.6)…”

L158: “…than single-born kids (24.2%).”

L159: “presented” instead of “given”

L160: “…variance components. It appears a…”

L162-163: “the highest”

L164: Where are these results presented? According to Table 2, the range of correlations is different.

L181: “…reported by other studies [27,28,29]. Similarly, our ST h² estimate is in the range reported by previous authors [30,31].”

L182: “Taking into consideration” instead of “Given”

L183: “considered as” instead of “thought to be”

L183-184: “…in the current situation, because the expected…”

L185: “However” instead of “But”

L185: “higher standards of” instead of “better”

L187: “was” instead of “is”

L193: “could” instead of “may”

L195: “concern is the” instead of “element concerns”

L204: “…should not have to be neglected.”

L206: “performed” instead of “done”

L207: “…high suggesting that these…”

L208-209: “As it has also been reported in previous studies [43,22]…”

L210: “measurements” instead of “measure”

L213: “Estimates of repeatability for BW, CG, BL and HW measurements across ages were all around 30%.”

L214: “Concerning the consistency of these measures, it is indicated that…”

L215: “with accuracy” instead of “accurately”

L215-216: Please rephrase

L219: “since” instead of “as”

L222: “destroyed” instead of “decimated”

L222-223: “…that animal diseases are responsible for about 18% of livestock deaths in…”

L223: “At institutional level, Ilukor [47] proposes…”

L224: “…among which the most important are the role of FFS and…”

L229: “…program could be accompanied…”

L234: “Results of estimates for the examined genetic parameters of this study…”

L235-236: “…goats with higher weight and better conformation.”

Author Response

(The authors gave the same response as above.)
